# Estimating the Probability Distribution of Construction Project Completion Times Based on Drum-Buffer-Rope Theory

**Xun Liu** [1,*] **, Le Shen** [2] **and Kun Zhang** [3]

1    School of Civil Engineering, Suzhou University of Science and Technology, Suzhou 215000, China
2    School of Business, Suzhou University of Science and Technology, Suzhou 215000, China; s793115763@163.com
3    Institute of Engineering Management, Hohai University, Nanjing 211100, China; dreamerzk@126.com
*    Correspondence: liuxun8127@usts.edu.cn; Tel.: +86-136-7510-2267

**Abstract:** Various factors affecting the construction progress are regarded as bottlenecks giving rise to the project duration overrun. The contractor should combine the project schedule with the plan in order to reduce the uncertainty of the project activities. The present research describes the method derived from the theory of constraints (TOC) attempts to enhance the relationship among activities, to revise and further reduce the uncertainty of construction activities to improve the reliability of project progress. The elements of drum, buffer and rope (DBR) in TOC are added to PERT network schedule; through the identification of schedule in the bottleneck process, the implementation plan of the bottleneck is obtained. By measuring buffer time and calculating network schedule buffer time as well as feeding time, the relationship among activities and uncertainty of duration are also improved. To illustrate the impact of DBR applications on improving project schedule reliability, a case of hydropower station as an example is illustrated to show enhanced reliability of scheduling. As compared to program evaluation and review technique network (PERT) simulation, the simulation results showed that the uncertainty of construction progress could be reduced if the DBR are well cooperated mutually.

**Keywords:** construction project; PERT; theory of constraint (TOC); drum-buffer-rope (DBR); construction schedule

## 1. Introduction and Literature Review

There are many uncertain factors in the construction process and they often have negative impacts on the project duration, resulting in project duration stipulated in the contract when the project plan does not match the duration of the practice. Just from the standpoint that the implementation process of the construction project requires a stable environment, if the difference between project completion and intended completion period is quite large, there may be several negative results like increasing idle time of intermediate task in construction process and negative influence on resource distribution rationality. Therefore, these uncertain factors would become bottlenecks to reduce project schedule uncertainty. However, owing to the correlation [1,2], transmission [3] and non-superposition [4] among these risk factors, project managers cannot fully take into account the impact of uncertain factors on activities [5]. In particular, the way the uncertainty is managed in the project and risk management havea direct influence on the success of a project. According to a previousreport, only 44% of the projects could catch up with the finishing line, while 70% of the projects reduce the anticipated work amount, and 30% of the projects were just simply terminated [6]. PERT is a traditional method for modeling uncertainties in project networks, in which the effect on project progress could be reduced through deriving from uncertainty based on buffer mechanism [7]. For example, project managers usually make activities duration under uncertainty influence joined to all levelsof process continued time, to ensure a single activity or overall project can be accomplished

within time as scheduled [8]. However, project managers are usually too conservative, or in order to ensure the protection engineering progress implemented as planned, the reliability of the project schedule is very low [7,8]. As a consequence, the higher uncertainty of the schedule, the more prejudice appropriate management measures like rational allocation and optimization of resources in the project [9]. Construction projects are usually executed with various resource constraints, which may change the critical activities of the project and change the project completion time [10]. Therefore, it is necessary to develop a technique that is capable of finding the critical activities and the project completion time by considering the activities' resource requirements and predecessors, and the uncertainties in their durations.

In terms of the view that the duration of every activity is stable, all kinds of risk factors that affect the schedule of the project should be regarded as the bottleneck in TOC [11,12]. Traditional PERT can weaken the probability characteristics of each activity that exists in project network [13]. Simulation-based scheduling could enhance the value of traditional scheduling methods by relaxing some of the restrictive assumptions of PERT [5]. The reliability could be enhanced by describing the project completion time as a probability distribution [14]. However, enough attention has not been paid to the normality assumption that is built into these scheduling methods, the opportunity to improve the reliability of these scheduling methods by finding the best fit probability distribution functions of the many activities in a schedule and an exact probability distribution function of project completion times is ignored [15].There is usually an underestimation of the true project meanby the PERT calculated mean project duration [7,13,16]. Moreover, the reliability of the implementation of the project PERT network progress planning is relatively low. Many researchers have investigated the project scheduling efforts, but it is usually hard to achieve their requirements [17–19]. Due to the uncertainty of constraints, the real concern in construction site for a contractor is how to manage the variation of project duration without reducing it [7].

The uncertainty of a project is the key issue and the potential main cause for most problems [20,21]. In response to the bottleneck caused by the uncertainties, project managers often actively increase the buffer time to absorb the delay of activity duration caused by the occurrence of these uncertainties during the process of construction management [6,11]. The constraint theoryaims to solve the bottleneck management problem, and could thus improve the overall operations and achieve maximum benefits [22]. In Goldratt's opinion, the most important factor is that itcan make actual progress, not conform to the planning progress, which is also related to the bottleneck of the project. Later on, he developed the "drum-buffer-rope" (DBR) schedule planning and control techniques using TOC [23]. Production management uses the DBR principle to rectify the uncertainty of activities and enhance the reliability of schedule. Goldratt indicated that the uncertainty of bottleneck activities duration can be improved by the element "drum", and the originally independent activities can be changed into the correlation mutually by the two elements "buffer" and "rope". The application of the three elements "drum", "buffer" and "rope" in the project schedule can make a contribution to reduce the uncertainty of the construction plan.

Compared with the progress management effects of the traditional industrial manufacturing process, JIT system, Cook indicated that the uncertainty can be significantly reduced after using TOC in progress management [24]. Blackstone also pointed out that with Ford Motor Company in the United States, the on-schedule delivery time achievement rate rose after applying the TOC [25]. Gardiner [26], Spencer, Cox [27] and Wu [28] thought that the application of the element "rope" can help to determine the start time of the bottleneck activities, which ensure that predecessor activities can be completed on time through the settings that project resources are totally put into the bottleneck activities and the start time of bottleneck activities are the same as scheduled.Steyn believed that the application of the TOC can harmonize the relationship between risk factors and the project plan organization, and can be more effective in reducing the uncertainty of the project construction schedule [29]. In view of the fact that the buffer mode cannot effectively guarantee the

project completed on schedule in some projects, Vonder et al., proposed a distributed buffer setting mode, and thought that the buffer mode had good robustness with the premise of completion on time [30]. Hu et al. [31] described the three components: plan, control and concentration of the DBR management scheduling model method in detail, and pointed out that such a management method is suitable for resources scheduling of construction projects. Zhang et al., proposed the buffer setting method under uncertain project conditions in using TOC [32] that includes several factors such as resource strain, network planning complexity and risk preference of project managers. It solved the problem that resource strain is difficult to be quantified and unified, and also took into account the using of alternative resources to solve the problem of resource shortage. In consideration of the characteristics that there are gradual gaps in activities and many uses of the element "drum" in the project plan, Bie et al., analyzed the weakness of the centralized buffer setting method in coping with some "drum" elements of the activities network, and proposed one setting method for dispersed capacity constraints. Additionally, he also obtained the project duration, which can be reduced to a great extent under the setting method for dispersed capacity constraint through experiments in different networks with "drum" elements [33]. Apparently, the element "drum" of DBR progress management and control technology in TOC can improve the uncertainty of bottleneck operation duration. The two elements "buffer" and "rope" can strengthen the correlation among the operations activities, which is originally independent. As a result, the application of the three types of elements "drum", "buffer" and "rope", combined with traditional PERT progress management technology, can effectively reduce uncertainty of the project construction schedule [15,26,29].

On the basis of previous research, this paper applied the "drum-buffer-rope" construction schedule management and control technology into a PERT network for improving the relationship among activities. The ultimate goal is to reduce the uncertainty of construction project schedules. The present research (1) proposed simulation model under uncertainty duration, which is founded on the definition that the completion period of the project is within the scope of the contract, and the time distribution for the project should be reduced; (2) the DBR in TOC is applied in construction project scheduling and control; (3) combined with the Monte Carlo simulation method to set up new project construction schedule to reduce the uncertainty of the construction project;and (4) thescheduling of a concrete rock fill dam project was applied as a case study to demonstrate and verify the validity of the proposed model.

## 2. Theory of Constraints

The TOC concept was addressed by Goldratt et al. [34] based on the principle that complex systems exhibit inherited simplicity. The constraints limit the system's ability to generate the system's real goal. TOC aims to increase throughput while simultaneously decreasing inventory and operation cost [27,34]. One of the thinking tools of the TOC is the effect–cause–effect. That is to say, there is a problem for which a cause is hypothesized. If the cause exists in reality, there are other effects one can predict. If the effect is found, the hypothesis will be strengthened. The five main steps of the TOC [23,35] are as follows: (1) identify the system's constraints; (2) exploit the identified constraints; (3) subordinate everything to the identified constraints; (4) make sure that the constraints are worked to the maximum; and (5) if in the previous steps a constraint has been broken, go back to Step 1.

Step (1) and step (4) are critical for an enterprise to apply the TOC methodology successively [36]. The main TOC technique to identify and exploit the constraint resources is named DBR [37]. As revealed in Figure 1, there are three types of buffers used in the DBR [38]: a constraint buffer, an assembly buffer and a shipping buffer. A constraint buffer is used to protect the schedule of the constraint and is inserted just before the choke point. An assembly buffer is used to ensure that parts coming from a constraint resource do not have to wait for parts coming from non-constraints, and it is located in front of an assembly operation that is fed by both constraint and non-constraint parts. A shipping buffer is used to protect the delivery dates of the orders and is, therefore, located at the end of the process.

That is to say, both the constraint buffer and the assembly buffer are closely related to the constraint machine.

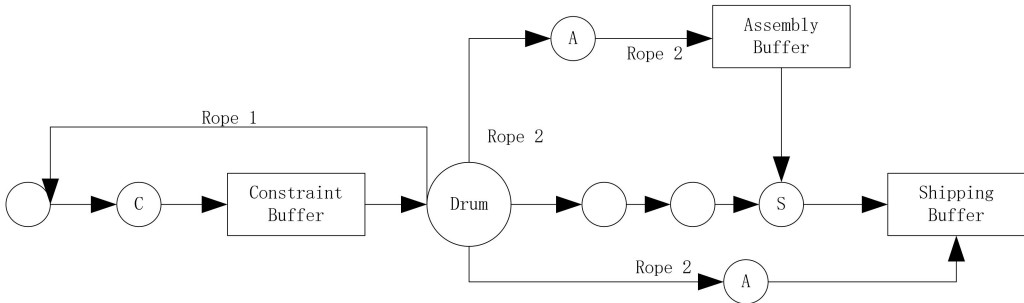

**Figure 1.** Drum–buffer–rope.

In Figure 1, "Drum" is a control point in the production system associated with the constraint (bottleneck) and its schedule. "Buffer" is time or a time equivalent amount of work in the process, and includes the constraint buffer and the shipping buffer. "Rope" is the term used for the communication feedback to the resources in front of the constraint resources so that each of them produces only the amount that the bottlenecks can complete. Based on this feedback, the entire production of the plan is based on the capability of the bottleneck. In other words, a maximum limit on the number of activities released to the bottleneck but not yet completed is established, and an activity is released whenever the number of activities is below the limit. There are two ropes: Rope 1 determines the schedule at the bottleneck to exploit the constraint according to the organization's goal; Rope 2 then subordinates the system to the bottleneck activity.

The central part of the DBR technique involves inserting buffers in front of the constraint resources and assembly operations to protect the production system from the inevitable fluctuations that are usually caused by the internal disruptions that occur in production of processing time. Contrary to MRP and JIT, in TOC the buffer is considered as a production strategic reserve that can protect the bottleneck from fluctuations in the production process.

### 3. Method
#### *3.1. Identification and Scheduling of the Bottleneck Operations*
#### 3.1.1. Identification the Bottleneck Operations

TOC technology is laid stress on confirming the bottleneck to make non-bottleneck operations fully cooperate with the whole production system [39], so that the whole production system can have a maximum producing capacity without changing the production flow time, and finally it can improve the practical effect of scheduling. Uncertainty on the critical path affects the improvement of the construction schedule. The bottleneck operation in PERT can be regarded as the most influential operation, resulting to the difference between actual and planning construction duration [36,40,41]. The greater the uncertainty or variance of process operation duration is given, the higher possibility that this operation can be a bottleneck in the schedule. On the other hand, the critical path can largely affect the uncertainty, which is also considered as the key chain and affected the reliability of project completion time. Consequently, there are four principles as follow to recognize the bottlenecks:

(1) Project duration is influenced by the major critical path of which the total time difference is zero or minimum. If duration on the critical path does not match the schedule, it makes the start time of subsequent activities and project completion time change.

(2) When the DBR schedule management technique is applied in the process of compiling a project schedule, since the bottleneck operations must limit operation, which reduces the certainty of project completion time, the bottleneck operation must exist

in the critical path. The degree of uncertain operational time can be used as the selection standard of bottleneck operations. In other words, the greater the standard deviation of the operation time that is given, the higher possibility that it may be the bottleneck operation.

(3) Sometimes the critical path is not only one, and each critical path may have a chance to have an effect on distribution of construction, in addition to considering the standard deviation of operation duration, selecting the bottleneck operations also needs to take into account the standard deviation of critical path completion duration at the same time.

(4) When there are several bottlenecks in a project network, the impact of the operation near the convergence point is higher than other operations, and the operation closest to the convergence point should be chosen.

### 3.1.2. Scheduling of the Bottleneck Operations

Goldratt [23] and Xie et al. [42] pointed out that the drum is determined by backward scheduling from customer orders. Other activity schedules obtain the expected duration of the total processing time from the drum scheduling [29,30]. Since the production schedule is subject to bottleneck, scheduling managers should have to provide sufficient resources including human, machinery and construction materialsin order to ensure the bottleneck activity can be started at the expected time and completed within the scheduled time. Therefore, there should be enough resources in bottleneck activities to reduce the uncertainty of bottleneck activity completion time caused by internal and external risk factors. As a result, activity schedules of bottlenecks should be the most possible operation completion time.

### 3.2. Determine Buffer Time of Bottleneck Activities

The differences among different projects should be considered. For example, project risk, owners and contractors, and managers often increase duration of activities subjectively. This increased time is named as buffer time or protection time, andis used to prevent the occurrence of activity time uncertainty. However, the activities actually do not have clear and stable buffer times; it means that the buffer times are different because of different characteristics of specific activities.

Usually, duration estimates for individual activities contain some arrangements for possible events or occurrences. DBR scheduling technique considered all possible events or occurrences into a project buffer. This implies that all expected times on individual activities and sub-projects are estimated. Buffers, on the other hand, are calculated to reflect the uncertainty in the estimates of duration of activities [43], and there is no exact constant rate of buffer time in construction activity. For example, the setting out process in surveying engineering may only need half a day sometimes, but floor concrete pouring operation needs to have 14 days for a maintenance period. For each activity to have the same rate of buffer time is simply not practical. If each activity has the same rate of buffer time, it would not match the actual construction situationand notaffect the quality and safety of construction project. This study adopts dynamic buffer, the bigger the variance is, the more the buffer is required. Additionally, the buffer time will be relatively decreased, when the activity completion time falls behind the expected schedule [44].

One of the challenges in DBR is the sufficient sizing of the buffers. If the buffers are estimated more than the necessary size, practical consequences immediately occur. Contrarily, if the buffers are underestimated, they may increase the probability of duration overruns, which can cause financial penalties and a reliable loss on the part of the customers or market. In study of the application of TOC in construction project schedule management, scholars have tried different methods to determine the buffer time of operations. Slusarczyk et al., attempted to apply these concepts and explored the advantages of applying TOC to a complex mega infrastructure project and to compute the buffer size using some of the available methods [45]. In a previous study, software development projects for resource-constrained

problems were analyzed and given solutions; an improved root square error was suggested; the setting method of buffer sizes, which is suitable for software development projects, was adopted; and the preemptive scheduling method based on a heuristic algorithm and priority rules was used to plan the scheduling [42]. A buffer sizing method based on comprehensive resource tightness was proposed to better reflect the relationship among activities and to improve the accuracy of project buffer determination [44]. Wei et al., considered the inline mode of security time for each resource conflict activity and proposed to set a reduction ratio to improve the calculation of the buffer time [46]. On the other hand, in the field of scheduling of construction projects, the determination of buffer time was not absolute. For example, Schragengein thought the size of buffersshould be three times the standard deviation of the average bottleneck lead time. He used three times based on relevant work experiences and assumed the reliable lead time complies with the normal distribution [47]. Ronen and Starr thought that the buffer time should be a quarter of the total lead time [48]. Cohen thought that it should accord to the degree of uncertainty level of the target or the sum of 50% of each activity duration [49]. At the same time, the buffer time can be regarded as degree of uncertainty of the activity duration. The higher the degree is, the longer the buffer time is, and also the more possibility that the project duration meet the progress schedule.

Schragenheim [47] and Demmy [44] pointed out three buffer times in manufacturing system constraint buffer, assembly buffer and shipping buffer. The buffer time would be reduced if the operation completion time exceeds predetermined. If the activities can be done as expected and made no idle time in the following activities, it would be helpful to the project managers to arrange as well as the capital input and use of project resources. Additionally, on the other hand, the occurrence of risk has probably a negative impact on the completion as expected of the project. Therefore, impact caused by risk exists in between pessimistic completion time and expected completion time of the activity. Therefore, the buffer time can be calculated according to the following Equations (1)–(3).

$$CB_c = \frac{T_{cb} - T_{ce}}{2}, c \in S_{CB} \tag{1}$$

$$AB_a = \frac{T_{ab} - T_{ae}}{2}, a \in S_{AB} \tag{2}$$

$$SB_s = \frac{T_{sb} - T_{se}}{2}, a \in S_{SB} \tag{3}$$

where

$CB_c$ = Constraint buffer (CB) time;
$AB_a$ = Assembly buffer (AB) time;
$SB_s$ = Shipping buffer (SB) time;
$T_{cb}$ = Pessimistic duration all predecessor activities of constraint buffer;
$T_{ce}$ = Expected duration of all predecessor activities of constraint buffer;
$T_{ab}$ = Pessimistic duration of all predecessor activities of assembly buffer;
$T_{ae}$ = Expected duration of all predecessor activities of assembly buffer;
$T_{sb}$ = Pessimistic duration of all predecessor activities of shipping buffer;
$T_{se}$ = Expected duration of all predecessor activities of shipping buffer;
$S_{CB}$ = Set of predecessor activities of constraint buffer;
$S_{AB}$ = Set of predecessor activities assembly buffer;
$S_{SB}$ = Set of predecessor activities shipping buffer.

Constraint buffer (CB) is a set before the bottleneck operation activities. The purpose is to provide a protective effect produced by the bottleneck, so that the bottleneck resource can reach the goal of predetermined output in terms of progress schedule.The bottleneck buffer must be placed in front of the critical path activity to minimize the resource limit and maximize the duration reliability [15]. Assembly buffer (AB) ensures that the bottleneck is not delayed by postponement of other activities when it is formed by joining components

together. The PERT and CPM have a confluence of activities that give rise to duration variation, resulting in an increase in project duration uncertainty. AB is therefore added to the assembly node where the bottleneck and non-bottleneck are merged. Shipping buffer (SB) is established in the product shipping area and aimed to protect production to satisfy the order delivery date, in order to prevent the influence of uncertain factors in the production process that may delay the project delivery date. The traditional production process in the CPM or PERT scheduling belongs to the push system. Once the bottleneck duration is extended, the postponed duration causes a breach of the contract, which is why the shipping buffer occurs after a bottleneck.

The measurements of three buffer times are similar, but have different numbers of predecessor activities of bottleneck, so the buffer time varies. Equations (4)–(6) distribute each activity buffer time in accordance with the ratios of expected duration of predecessor activity of the buffer to expected project duration.

$$BT_c = \frac{b_c}{T_e} \times CB_c, c \in S_{CB} \tag{4}$$

$$BT_a = \frac{b_a}{T_e} \times AB_a, a \in S_{AB} \tag{5}$$

$$BT_s = \frac{b_s}{T_e} \times SB_S, s \in S_{SB} \tag{6}$$

where

$BT_c$ = Buffer time of activity $c$ before the bottleneck buffer;
$BT_a$ = Buffer time of activity $a$ before assembly buffer;
$BT_s$ = Buffer time of predecessor activity $s$ before shipping buffer;
$T_e$ = Expected project duration;
$b_c$ = Pessimistic duration of predecessor activity $c$ before constraint buffer;
$b_a$ = Pessimistic duration of predecessor activity $a$ before assembly buffer;
$b_s$ = Pessimistic duration of predecessor activities of $s$ before shipping buffer.

Given that buffer time is derived from each activity, the redundant safety protection time in each activity is removed to set buffers and centralized as a mechanism to deal with uncertain factors in project implementation process. Equations (7)–(9) remove extra buffer time from each activity. The removed buffer time is then placed in the buffer zone, and can be used to manage the project uncertainty and improve the scheduling reliability.

$$\overline{b_c} = b_c - BT_c \tag{7}$$

$$\overline{b_a} = b_a - BT_a \tag{8}$$

$$\overline{b_s} = b_s - BT_s \tag{9}$$

where

$\overline{b_c}$ = Pessimistic duration without buffer time of predecessor activity $c$ before CB;
$\overline{b_a}$ = Pessimistic duration without buffer time of predecessor activity $a$ before AB;
$\overline{b_s}$ = Pessimistic duration without buffer time of predecessor activity $s$ before SB.

### 3.3. Constraint Buffer Management of Activities

The settings of buffer time are commonly provided by each activity to remove redundant safety protection time in every activity, which is a centralized mechanism to deal with uncertain factors in the project implementation process [38,43]. Furthermore, project managers can monitor the project progress status and reduce the uncertainty through buffer management.

Nowadays, large projects have complex construction conditions including information, task, techniques, organization, environment, and goal, which determines the dynamic, uncertain and highly interdependent features of the project construction process and

system [20,50]. Immediate management of project duration from that fact is of extreme important. Schragenheim suggested that the length of buffer time can be divided into three sections: negligible zone, alert zone and accelerative zone, the length of each zone is equal [47]. The size of each zone is allocated equally according to Equations (1)–(3). The constraint buffer is shown in the following Equation (10).

$$\overline{T_{ce}} = \max\left\{ t_i + \frac{a_i + 4m_i + \overline{b_c}}{6} \right\} \tag{10}$$

$\overline{T_{ce}}$ = Expected duration of all predecessor activities of CB without the buffer time;
$t_i$ = Start time of activity $i$;
$a_i$ = Optimistic duration of activity $i$;
$m_i$ = Most possible duration of activity $i$;
$\overline{b_c}$ = Pessimistic duration without buffer time of predecessor activity $c$ before constraint time.

If the actual duration of the predecessor activity of the constraint buffer is longer than that of the expected duration, the project duration is not likely to meet the requirements, therefore, the constraint buffer needs sufficient time to absorb the extra project duration caused by uncertainties.

If the actual duration of the predecessor activity of the constraint buffer is within the negligible section, the buffer time is still sufficient for the project manager to make use of the project duration. If it is located within the alert section, the project manager should watch more closely on the progression of project duration. If it is within the accelerative section or alert section, the project can be expected to complete smoothly. Given the starting time of the bottleneck of adding $2 \times$ (CB/3), the temporal variances are in a range still acceptable for the project manager, thus posing no need of extra overworking resource to start the bottleneck activity ahead of the schedule, since it is more important for the manager keeping the project duration as planned than decreasing it.

While if the actual duration is within the accelerative section, the manager has to increase adequate resources, such as manpower, machinery for construction or working hours, so as to start activities ahead of schedule for earlier completion, allowing bottleneck activity to set in motion as scheduled. When confronting with existing possibility problems of penalty for breach of contract and project delay, it is explicitly necessary to add in a certain amount of resources to shorten the activity time, and that, in the meantime, adds costs to the project [51].

In the field of industrial engineering, the output pace of the bottleneck determines the output efficiency of the system [52]. On the other hand, if the materials were produced too late, activities would delay, which may also affect the arrangement of start time for bottleneck activities, and then influence the output of the whole construction system [51,52]. Therefore, the completion time of activities before bottleneck must be decided appropriately. The element rope was defined as the opposite length of lead time from bottleneck to order starting by Gardiner [26]. The element can make the production speed of all activities in the production system executed according to the production rhythm of the bottleneck. The control function of rope can be attained through the establishment of detailed plan from resources to construction site [26]. Wu thought that the main function of rope is to determine the proper time and correct materials arriving at the construction site [28]. Spencer explained that the purpose of rope control is to ensure the production materials are always enough to support the ongoing bottleneck activities. It means that the rope is used to decide the start time to book materials and resources in addition to make them smoothly go through the non-bottleneck activities and ensure the bottleneck activities to be completed on schedule [27].

It can be summarized that the predecessor activity duration of the bottleneck can be regarded as the scale of rope. The length of the rope is the same as the duration of the predecessor activity of the bottleneck, which uses a push system for synchronous production. The rope scheduling is the milepost for all bottleneck predecessor activities after the start

of work. To enable the bottleneck to start as expected in the buffer management, the total completion time of all predecessor activities must not exceed the rope scheduling [26,28,42]. When a constraint is broken, the next constraint needs to be identified and improved.

### 3.4. Simulation of Construction Project Scheduling in DBR Model

The method proposed in the study is described in the flowchart presented in Figure 2. The DBR model components include the drum, buffer and rope. For example, the schedule model simulations contain two schedule components (C = 2), three possible types of DBR model, the Drum–Buffer schedule, Drum–Rope schedule and Buffer–Rope schedule, which were compared with PERT simulation results. Therefore, the algorithm runs many simulation experiments using activity durations with the eight-schedule model.

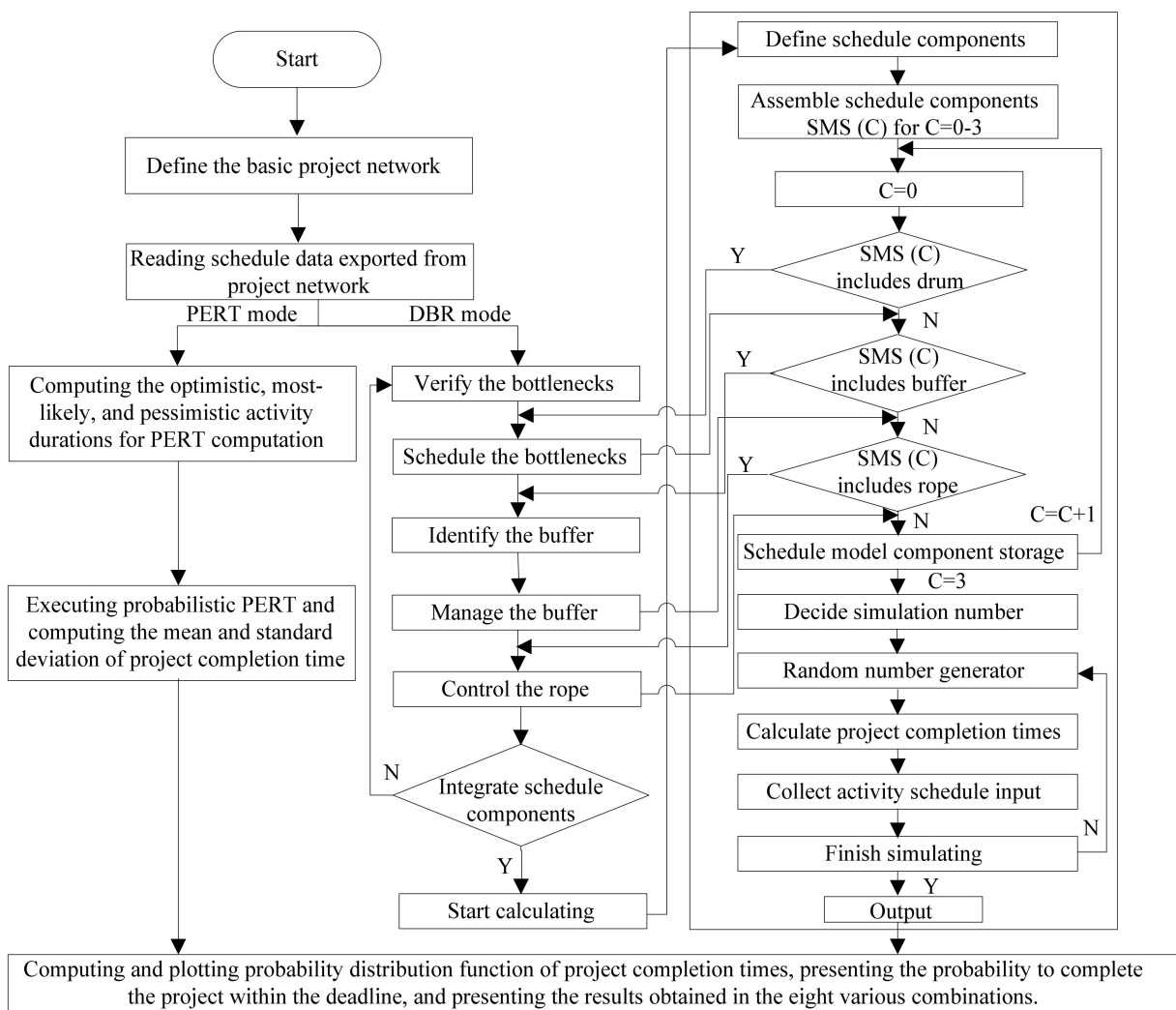

**Figure 2.** Flowchart of DBR model applied in construction project scheduling.

## 4. Empirical Examples

### 4.1. Background

A hydropower station is located on the main stream of the Yellow River, which is a large-scale cascade hydropower station. It is one of China's important power points in the northern channel of the Power Transmission from West to East in the Western Development Strategy. The reservoir design capacity of the hydropower station is 62 million cubic meters, which is a daily regulation reservoir. The main task of this power station is to generate electricity, in addition to the functions of irrigation and water supply. Main

hydraulic structures are made up of a concrete face rock fill dam, left bank flood discharging tunnel, left bank spillway, right bank flood discharging tunnel, water diversion and power generation system and powerhouse, with a total investment of 6.6 billion Yuan.

According to the engineering characteristics and the technological requirements of a concrete face rock fill dam and the actual situation of the hydropower station construction, after careful analysis and research, the owner, supervisor, designer and construction contractor reached a consensus that the deployment of the material resources, the arrangement of the construction road and the arrangement of building the dam are the three key factors to ensure the whole of the hydropower station project completed on schedule. For the filling and building construction of the concrete faced rock fill dam, a careful analysis of the relationship between the construction process activities, and an accurate quantitative expression of the correlations, dependencies and constraints among the process activities, are the prerequisite to make rational arrangement of the construction sequence. Additionally, on the foundation of this analysis, the key process and critical path obtained from the network and the problems is reflected, which has its practical guiding significance for managers and construction organizers to better understand the key of construction and to distinguish the primary and secondary points of works, in order to better achieve the project schedule control target.

### 4.2. Case Analysis

In the present research, a case study was brought to demonstrate the application of conjunctive use of the three elements drum–buffer–rope in practical project management, based on the fundamental hypothesis [53,54], the study in this part is used to show whether the method can effectively reduce the degree of uncertainty of the project construction period. The case in this study is analyzed as follows:

Step 1. Define the basic project network plan

Referring to the logical relations among processes specified in the overall progress of the construction network plan of the concrete panel rock fill dam of the hydropower station project (contract stage research report), the network plan of the general construction schedule can be drawn as shown in Figure 3. As the concrete panel rock fill dam has a long construction period, many processes and complicated relationships are associated with the processes, when drawing the network plan of the overall construction progress, digital as a code name was used for a different construction working procedure. Additionally, complete details of the process (process name, process activity continued time, process engineering quantity, antecedent process and subsequent process), triangular estimate value, expected completion time, variance, and standard deviation of each activity in project network are listed in Table 1.

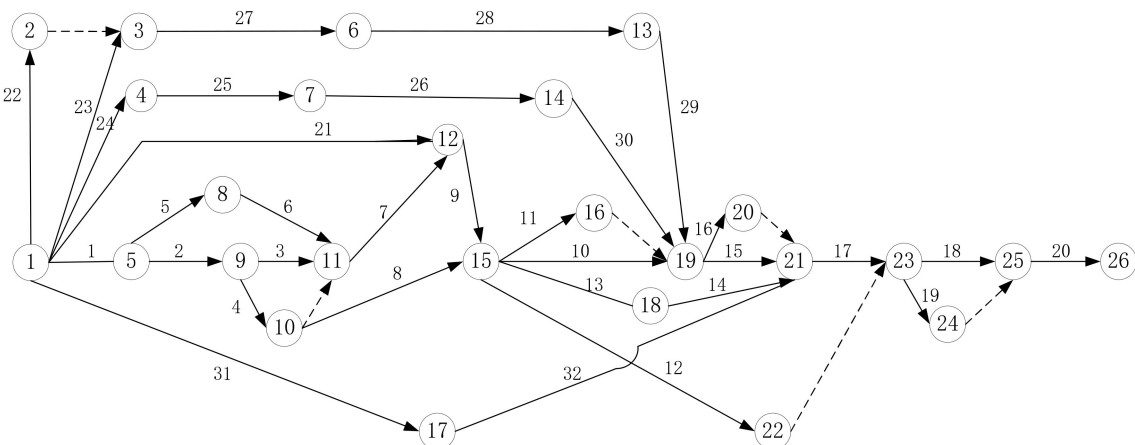

**Figure 3.** Double code network diagram of the construction of a concrete face rock fill dam.

**Table 1.** Detail table of engineering construction process information.

| Number | Activity Item | Precedence Relation | Duration Estimation (a, m, b)/Day | Activity Expected Duration | Standard Deviation (Day) | Variance |
|---|---|---|---|---|---|---|
| 1 | Excavation of dam abutment above water on two banks (above ▽1901) | - | 170,230,260 | 225 | 15 | 225 |
| 2 | Closure dike filling and foundation pit drainage | 1 | 40,55,70 | 55 | 5 | 25 |
| 3 | Foundation pit excavation | 2 | 50,70,90 | 70 | 6.67 | 44.44 |
| 4 | Foundation excavation of cutoff wall beside dam | 2 | 30,40,55 | 40.83 | 4.17 | 17.36 |
| 5 | Seepage construction of upstream and downstream enclosing wall | 1 | 40,55,60 | 53.33 | 3.33 | 11.11 |
| 6 | Filling construction of upstream and downstream cofferdam | 5 | 30,45,50 | 43.33 | 3.33 | 11.11 |
| 7 | Silicon pouring of toe slab and foundation (below ▽1901) | 3,4,6 | 8,15,25 | 15.50 | 2.83 | 8.03 |
| 8 | Silicon pouring of cutoff wall beside dam | 4 | 55,70,90 | 70.83 | 5.83 | 34.03 |
| 9 | Filling construction of temporary section of upstream dam (below ▽1955) | 7,21 | 140,165,185 | 164.17 | 7.5 | 56.25 |
| 10 | Filling construction of temporary section of down-stream dam (below ▽1955) | 8,9 | 130,150,185 | 152.5 | 9.17 | 84.03 |
| 11 | Backfilling of gully at axis on left bank | 8,9 | 90,120,145 | 119.17 | 9.17 | 84.03 |
| 12 | Masonry beside dam | 8,9 | 455,495,560 | 499.17 | 17.5 | 306.25 |
| 13 | Silicon pouring of panel of first stage | 8,9 | 45,60,80 | 60.83 | 5.83 | 34.03 |
| 14 | Water stopping installation on surface | 13 | 45,60,75 | 60 | 5 | 25 |
| 15 | Whole section filling from▽1955 to dam crest | 10,11 29,30 | 160,195,215 | 192.5 | 9.17 | 84.03 |
| 16 | Filling of slope body in front of dam (below ▽1940) | 10,11 29,30 | 100,120,145 | 120.83 | 7.5 | 56.25 |
| 17 | Silicon pouring of panel of second stage | 14,1516,32 | 75,90,115 | 91.67 | 6.67 | 44.44 |
| 18 | Construction of parapet wall and road on dam crest | 12,17 | 85,120,150 | 119.17 | 10.83 | 117.36 |
| 19 | Demolition of downstream cofferdam | 12,17 | 70,90,115 | 90.83 | 7.5 | 56.25 |
| 20 | Filling and masonry on dam crest | 18,19 | 48,60,75 | 60.5 | 4.5 | 20.25 |
| 21 | Bolt-concrete support of dam abutment on two banks | - | 280,330,400 | 333.33 | 20 | 400 |
| 22 | Construction of helper system in this contract section | - | 135,165,195 | 165 | 10 | 100 |
| 23 | Transformation of machining system of cushion material | - | 85,105,115 | 103.33 | 5 | 25 |
| 24 | Borrow Area Planning and road construction in II zone of water ditch | - | 25,30,45 | 31.67 | 3.33 | 11.11 |
| 25 | Peeling of gravel soil and strong decomposed rock | 24 | 100,120,145 | 120.83 | 7.5 | 56.25 |
| 26 | Mining and blasting test of transition material | 25 | 12,15,20 | 15.33 | 1.33 | 1.78 |
| 27 | Machining of test material in cushion | 22,23 | 35,45,50 | 44.17 | 2.5 | 6.25 |
| 28 | Machining of cushion material in first stage | 27 | 48,60,80 | 61.33 | 5.33 | 28.44 |
| 29 | Machining of cushion material in second stage | 28 | 105,135,160 | 134.17 | 9.17 | 84.03 |
| 30 | Mining of transition material and cushion material | 26 | 195,240,280 | 239.17 | 14.17 | 200.69 |
| 31 | Grouting test | | 72,90,118 | 91.67 | 7.67 | 58.78 |
| 32 | Grouting and pouring of toe slab on left and right banks | 31 | 300,360,420 | 360 | 20 | 400 |

Note: in the table, a, m, b, respectively, indicate the optimistic completion time, the most likely completion time and the pessimistic completion time of the process. The value is obtained by modifying original data.

Step 2. Verify the bottlenecks

As shown in Table 1 and Figure 2, the activities on the critical path are 1, 2, 3, 7, 9, 10, 15, 17, 18 and 20. The critical activities are ranked from large to small according to their variation or standard deviation,1,18,10 or 15,9,3 or 17, 2, 20, and 7. As a result, activity 1, 2, 3, 7, 9, 10, 15, 17, 18 and 20 would be thebottleneck activities, and thus, the bottlenecks schedule and the buffer and the rope could be built up.

Step 3. Schedule the bottlenecks

In this paper, using the commercial software Crystal Ball Version, the probability duration distribution of each operation is set as a triangular distribution, which is composed with the most optimistic completion time, the most likely completion time and the most pessimistic completion time [55–57]. The calculations were based on the simulation flowchart shown in Figure 2. By setting a certain number of simulations, for example, setting simulation time up to 20,000, through a random number generator (0–1) and the triangular distribution operations, the completion time of each operation, the project completion time and probability distribution can be obtained. In the progress of planning the model building process, it is necessary to consider the selection and combination of schedule elements, calculate the required data of the schedule elements, and coordinate with the calculation of the PERT schedule. The DBR technical elements include three basic elements of drum, buffer, and rope. It has seven planning patterns in combination. With the traditional PERT network planning model, a total of eight schedule models exist. Through the comparison among the eight schedule models, a suitable project schedule can be found, so as to be able to reduce the completion risk to a minimum. After building the model, the schedule can be placed into the temporary storage. The number of Monte Carlo simulations and operation duration of each simulation can be set according to triangular distribution and random number generator. Once the operation duration of each simulation is determined, the completion time of the project is calculated according to critical path method, until the expected times of simulation are reached.

### 4.3. Results and Discussion

The simulation results under PERT network schedule of this case and the simulation results including various combinations of different elements under DBR scheduling technique is shown in Figure 4. The data comparison of a variety of simulation results obtained by different progress planning methods are shown in Table 2.

The expected duration in the project progress network diagram is 1211.63 days, but uncertainty of duration in the network schedule in terms of PERT schedule technique is as high as 33.56 days (SD. 33.56 days, Min. 1092.29 days, and Max. 348.92 days). Additionally, when the DBR technology is applied in PERT network diagram, compared to the traditional PERT network technology, drum technology reduces the expected project completion time by 2.68 percent (31.59 days), while the uncertainty of the project completion duration has decreased by 3.8 days (SD. 29.76 days, Min. 1083.51 days, Max. 1288.12 days). Application of buffer technology in projects increases the average completion time by 6.03% (73.01 days), but the degree of uncertainty of the project completion period has decreased by 16.78 days (SD. 16.78 days, Min. 1220.30 days, Max. 1344.39 days). Similarly, the application of rope technology also increases the average completion time of the project, and the rate of increasing was 1.27% (15.44 days), but the uncertainty of project completion period has decreased by 6.97 days (SD. 26.59 days, Min. 1132.12 days, Max. 1322.37 days). When putting two of schedule control elements into the traditional PERT project schedule network diagram, drum-buffer technology increases the expectations of the project completion time, the rate of increasing was 3.59 % (43.52 days), but uncertainty of project completion period has decreased by 22.16 days (SD. 11.40 days, Min. 1213.72 days, Max. 1303.21 days). Drum-rope technology reduces the average completion time of the project 1.27% (15.44 days), while the uncertainty of completion period reduces 10.45 days (SD. 23.11 days, Min. 1117.47 days, Max.1271.83 days). Buffer-rope technology reduces the average completion time of the project 1.91% (23.17 days), while the uncertainty of completion period reduces 24.38 days

(SD. 9.18 days, Min. 1152.89 days, Max. 1216.75 days). As for the comparison between drum-buffer-rope and traditional PERT schedule technology, the DBR reduces the average completion time of the project 2.6% (27.38 days), while the uncertainty of completion period reduces 25.64 days (SD. 7.92 days; Min. 1156.38 days; Max. 1212.46 days).

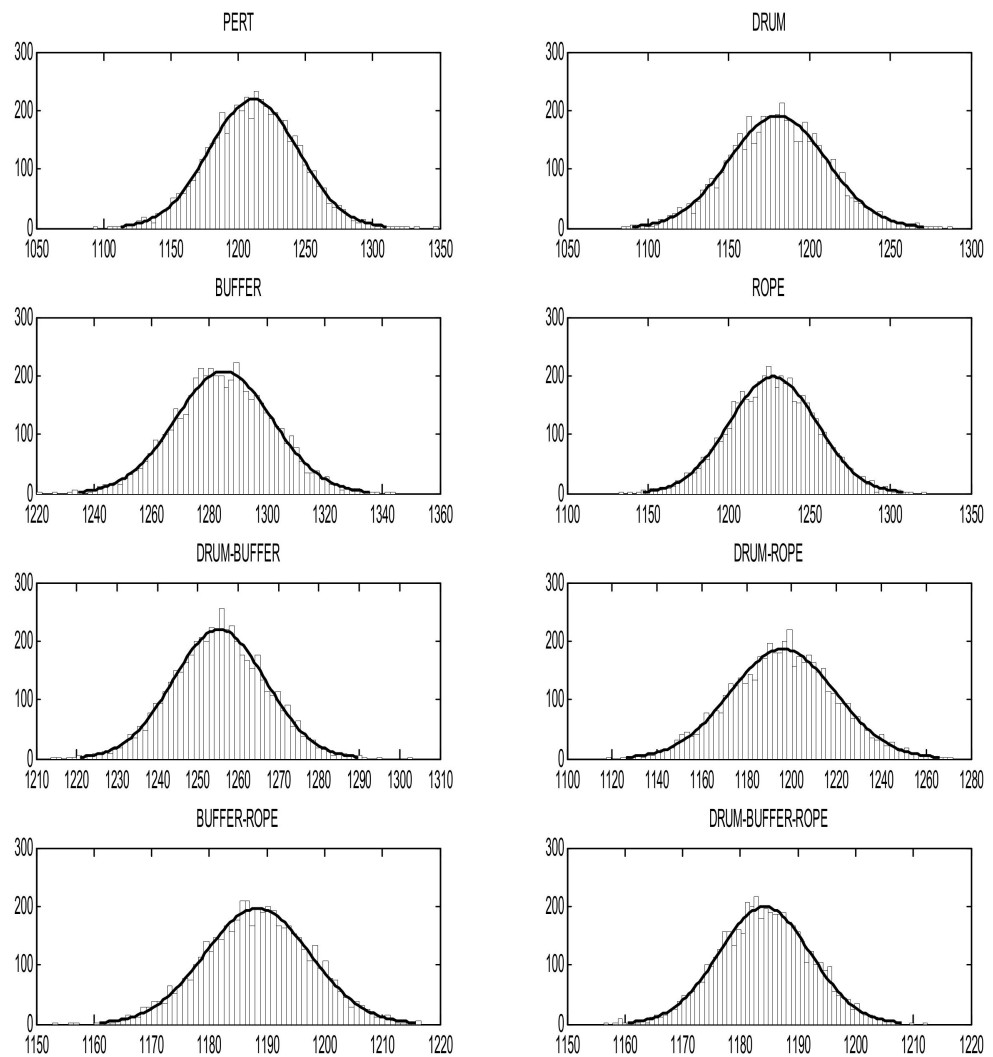

**Figure 4.** Simulation results under different combinations.

**Table 2.** Comparison table of simulation uncertainty of various combinations.

| Number | Control Elements of Progress Schedule | Average Completion Period | Standard Deviation | The Most Optimistic Completion Time | The Most Pessimistic Completion Time | Uncertainty Reducing Compared with PERT |
|---|---|---|---|---|---|---|
| 1 | PERT | 1211.63 | 33.56 | 1092.29 | 1348.92 | - |
| 2 | drum | 1180.04 | 29.76 | 1083.51 | 1288.12 | 3.8 |
| 3 | buffer | 1284.64 | 16.78 | 1220.30 | 1344.39 | 16.78 |
| 4 | rope | 1227.07 | 26.59 | 1132.12 | 1322.37 | 6.97 |
| 5 | drum-buffer | 1255.15 | 11.40 | 1213.72 | 1303.21 | 22.16 |
| 6 | drum-rope | 1196.19 | 23.11 | 1117.47 | 1271.83 | 10.45 |
| 7 | buffer-rope | 1188.46 | 9.18 | 1152.89 | 1216.75 | 24.38 |
| 8 | drum-buffer-rope | 1184.25 | 7.92 | 1156.38 | 1212.46 | 25.64 |

Observing the comparison chart of completion period distribution of various project schedule network simulation under different combinations (Figure 5), and comparison

chart of completion probability under different combinations (Figure 6), through comparison the results in Figures 5 and 6, it is clearly that although the average completion time in traditional PERT project scheduling techniques are not the longest completion time, the degree of uncertainty of completion time calculated in this method is the highest. The higher degree of uncertainty of completion duration, the more likely there is additional operation idle time, and the more likely the allocation of operation resources fluctuates greater. These all result in the failure of management. However, under the conditions of independent application of drum elements, buffer elements and rope elements, the uncertainty of the progress plan can be reduced to some extent compared with PERT technology. In other words, in the case that these elements are independently applied, the results obtained are able to improve the uncertainty distribution of project completion time. However, in reality, the actual situation is that the case is also likely to result in a great degree of the extension of the project completion time. It is not the best approach to reduce the uncertainty of project progress. Similarly, in the case of the mutual application of drum elements, buffer elements and rope elements, the combination technology can reduce the uncertainty of completion duration of the plan to some extent, but may also cause increasing of the project completion time. In the joint application of drum elements, buffer elements and rope elements, although it is not the best choice to reduce uncertainty of the project network plan, it does not increase the project completion time. Under the constraints of the uncertainty of the project schedule and project completion time, the schedule control technique that drum elements, buffer elements and rope elements are jointly applied can get more appropriate effects on project progress scheduling.

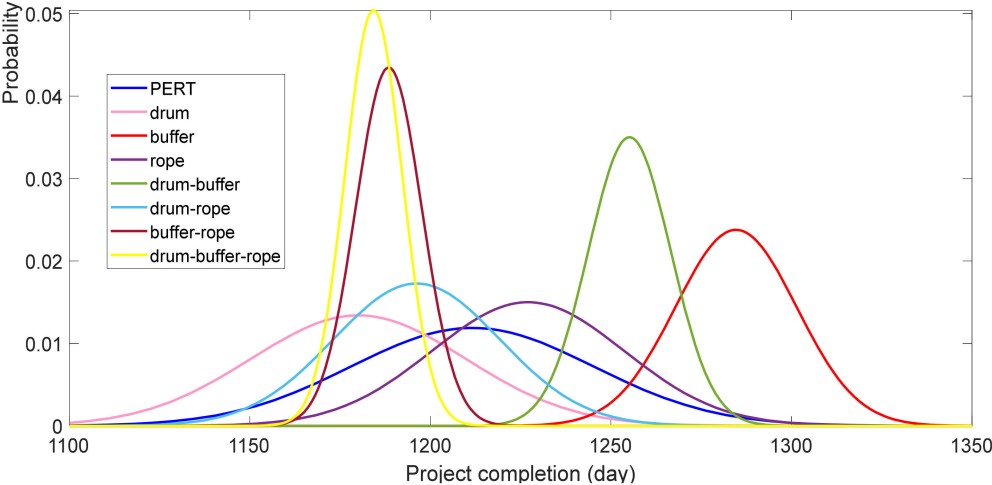

**Figure 5.** Simulation comparison of completion period distributions between different combinations.

As used herein, the management model of project progress proposed in this paper is adding each element of DBR scheduling and management technique in the TOC into the traditional PERT network schedule. The application of each element can effectively reduce uncertainty of completion period, and there is also a case study to demonstrate that the conjunctive use of three elements drum-buffer-rope can most effectively reduce the degree of uncertainty of the project construction period in this research.

This study also shows that when a progress schedule is made by using traditional PERT network plan, the application of drum elements can help project managers to decide the bottleneck process and arrange the plan of bottleneck process in order to achieve the purpose to reduce uncertainty of progress schedule. However, it still needs to be used in conjunction with the two elements of buffer and rope. Taking the buffer elements into account in progress scheduling process can help managers to remove redundant security protection time in each process, and make a centralized management as a mechanism to deal with the uncertain factors in the construction process of the project. Additionally,



managers also are able to monitor the executed state of the progress under the help of buffer management technology, in order to reduce the uncertainty of scheduling. Traditional progress management techniques, such as CPM and PERT techniques, calculate the operation activities starting time by a method of forward reasoning. However, the construction process of a project is a propulsion system; a delay on the critical path will result in the completion period not meeting the schedule plan. In other words, the uncertainty of the construction schedule will increase. If the use of rope elements can help to control the start time of project bottlenecks, the impact from subsequent operation duration on the uncertainty of completion period will be reduced. If the project schedule makers can make the three elements of drum–buffer–rope be fully fit when they are developing the progress schedule, it is possible to form a good DBR schedule and control technology, which can be effectively applied to the actual schedule control of the project.

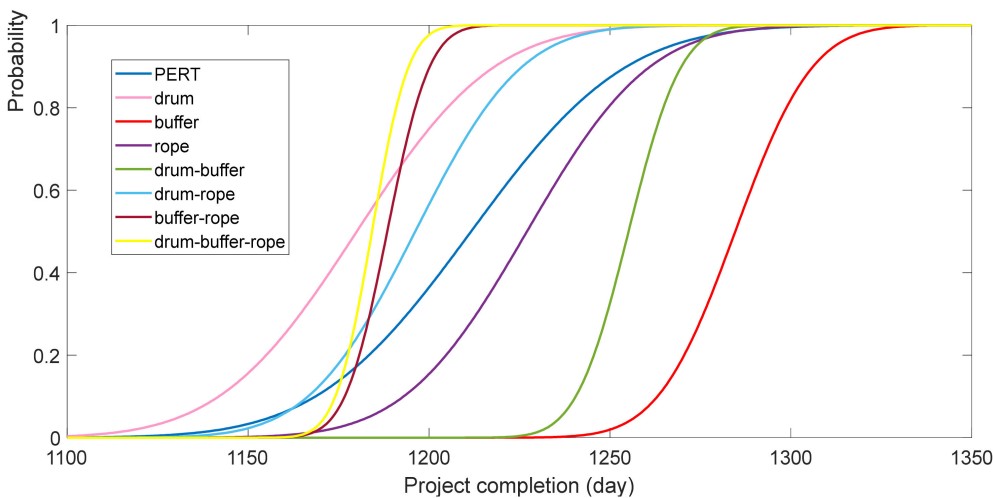

**Figure 6.** Simulation comparison of completion probability between different combinations.

## 5. Conclusions

The project schedule management model proposed in this study uses each element of the DBR schedule and management technology in the theory of constraints to join the formulation of the traditional PERT network schedule, and the application of each element can effectively reduce the uncertainty of the project completion period. At the same time, the calculation example also proves that the combined use of the three elements of Drum–Buffer–Rope can most effectively reduce the degree of uncertainty of the project completion period. The research in this study also shows that when using the traditional PERT network to prepare schedules, the use of Drum elements can help the project manager to determine the bottleneck process and arrange the schedule plan of the bottleneck process to reduce the inconsistency of the schedule and to obtain the degree of certainty, but this still requires the use of the two elements of Buffer and Rope; the Buffer element is taken into account in the formulation of the project schedule, which can remove the excess safety protection time in each process and concentrate it. Management is used as a mechanism to deal with uncertain elements in the construction process of a project. At the same time, it can monitor the execution status of the project schedule through the buffer management technology to achieve the purpose of reducing the uncertainty of the schedule; and the traditional project schedule management technology, such as CPM and PERT technology, the forward calculation method used when calculating the start time of the process activity, and the construction process of the process operation belongs to a propulsion system. Once the process on the key line is delayed, the project completion period will not meet the schedule plan. That is, the uncertainty of the construction schedule is increased. If the Rope element can be used to control the start time of the project bottleneck, the impact of the subsequent process operation duration on the uncertainty of the completion period will

be reduced. The three elements of Drum–Buffer–Rope can be fully coordinated when the schedule is formulated to form a good DBR schedule and control technology, and can be effectively applied to the actual schedule control of the project, so that the benefits of DBR's engineering project schedule and control technology can be more effectively revealed.

**Author Contributions:** Conceptualization, X.L. and L.S.; methodology, X.L. and L.S.; formal analysis, X.L. and K.Z.; investigation, X.L. and K.Z.; data curation, X.L.; writing—original draft preparation, L.S.; writing—review and editing, X.L.; supervision, X.L. All authors have read and agreed to the published version of the manuscript.

**Funding:** Philosophy and Social Science Research in Colleges and Universities in Jiangsu Province (No. 2020SJA1394), Fundamental Research Funds for the Central Universities (No. 331711105), Jiangsu Provincial Construction System Science and Technology Project of Housing and Urban and Rural Development Department (No. 2017ZD074).

**Institutional Review Board Statement:** Ethical review and approval were waived for this study, due to this study not involving biological human experiment and patient data, which was not within the scope of review by the Institutional Review Board of Suzhou University of Science and Technology.

**Informed Consent Statement:** Informed consent was obtained from all subjects involved in the study.

**Data Availability Statement:** The data presented in this study are available on request from the corresponding author.

**Acknowledgments:** The authors would like to thank the reviewers for all helpful comments, and to thank the foundation of Philosophy and Social Science Research in Colleges and Universities in Jiangsu Province (No. 2020SJA1394), Fundamental Research Funds for the Central Universities (No. 331711105), Jiangsu Provincial Construction System Science and Technology Project of Housing and Urban and Rural Development Department (No. 2017ZD074), Jiangsu Province Joint Education Program High-Standard Example Project, for their support.

**Conflicts of Interest:** The authors declare no conflict of interest.

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
