# Peer review of "Estimating the Probability Distribution of Construction Project Completion Times Based on Drum-Buffer-Rope Theory"

_applsci, doi:10.3390/app11157150_

Round 1

Reviewer 1 Report

All comments revolve around improving the bibliography except the last one (4):

  1. Authors should improve the introduction and review of the literature. More quotes are needed, as the full article is only 38. There are many statements in the introduction that are not accompanied by quotes to support such a phrase. Especially in the first two pages of the introduction. 
  2. Also in section "3.1.1. Identification the Bottleneck Operations" needs more bibliographic references. For example, the authors write about 4 principles to recognize bottlenecks. Have these principles been developed by the authors themselves in this article? If so, specify; otherwise, they should cite and respect authorship
  3. In section "4.1. Background" more bibliographic references are needed. 
  4. Is it possible to add the conclusions section? In the last two paragraphs of section "4.3. Results and discussion" the authors summarize the contributions of their research. It would be good to highlight such ideas in a "Conclusions" section.

Author Response

We appreciate the time and effort of the Reviewers and the Editor in reviewing our manuscript. The reviews are very helpful for us to improve the manuscript. As a result of the comments from both the Editor and the Reviewers we have made significant changes and have rewritten parts of the manuscript. Point to point respond to all comments are as follows. Revised parts are marked in RED color in the revised version.

Reviewer 2 Report

The authors deal with an interesting topic of construction project to   revise the uncertainty of activities and reduce the uncertainty of progress. Although the topic is very interesting, several major insufficiencies need to be improved.

Suggestions for improvement:

Introduction- The introduction is too long and a clear statement of the research purpose or establishing the research question is missing. Maybe some hypotheses should be assumed. It is also suggested to separate the sections Introduction and Literature Review.

Methods- the authors used the case study method, but they didn't quite explain why. Additionally literature about Case study methodology should be addressed.

In section 3. 2 The differences among different projects should be considered. For example, project risk, owners and contractors, managers often increase duration of activities subjectively based on conservative mentality. I don't quite understand what the authors meant. The mentality does not appear to be a factor that causes the duration of the action, the mental abbreviation used here is illegitimate, even considering that it is only an example.

Explanations of results and their discussion are clear, but the discussion about the research significance is missing. Also, add some additional discussion of findings in relation to the research framework as well as research goals and hypotheses are needed.

The authors are urged to draw conclusions that are more specific. At the moment it seems like good observations and arguments that are currently missing from the discussion section. There should be a clear connection with the research problem, goals, and results.

Results and Discussion- This section has to highlight the practical and theoretical implications of the results. These ones should be discussed in comparison with the findings in literature and the novelty of solution should be stressed.

In conclusion I recommend to the authors to improve the article according to the above mentioned issues.

Author Response

(The authors gave the same response as above.)
